# Cost Analysis of Collection and Ex Situ Conservation of Seeds of Important Native Tree Species of Mexico

Joel Rodríguez-Zúñiga [1], Manuel de J. González-Guillén [2], Horacio Bautista-Santos [1,3,*] and Fabiola Sánchez-Galván [3]

1   Tecnológico Nacional de México/ITS de Chicontepec, Barrio Dos Caminos Número 22 Colonia Barrio Dos Caminos, Chicontepec 92709, Veracruz C.P., Mexico
2   Colegio de Posgraduados, Posgrado en Ciencias Forestales, Montecillo, Texcoco 56264, Estado de México C.P., Mexico
3   Tecnológico Nacional de México/ITS de Tantoyuca, Desviación Lindero Tametate S/N Colonia La Morita, Tantoyuca 92100, Veracruz C.P., Mexico
*   Correspondence: horacio.bautista@itsta.edu.mx

**Abstract:** In Mexico, there are barely any studies that estimate the cost of germplasm conservation especially that carried out ex situ. The objective of this study was to analyze the collection and storage costs (CSC) of seeds of some native tree species that have social and economic importance in Mexico. Information on the importance of native trees was obtained through a literature review and research from a seed bank and forest technical service providers. Based on the environmental and social importance, as well as the conservation status of native species, poverty attention, and food security, an indicator of economic and social importance has been designed using multicriteria analysis. The highest value of the indicator included 32 species; Fabaceae was the most represented family (28%). The CSC analysis was applied to the species with the highest value indicator, currently available market price (CMP), and stored in the seed bank (SB). They were evaluated using the "cost of collection" method comparing CMP vs. SB. The cost of accession in the SB is 11 times higher than the CMP with 7% of nominal interest rate (NIR) and increases 24 times during a storage period of ten years with the same NIR per year. Despite the costs associated with the maintenance and management of SBs, their importance for conservation and contribution to society is highlighted.

**Keywords:** seed banks; poverty attention; economic and social importance indicator; food security

## 1. Introduction

Mexico is considered a megadiverse country, especially in terms of floristic richness. It is the fourth country in the world with the greatest floristic richness [1,2] with 22,314 plant species. Specifically, there are 2885 native tree species recorded, with high endemism of 44%, which makes Mexican flora unique [3].

The high biodiversity of tree species distributed in Mexico represents a valuable genetic resources source [4]. The unique characteristics of the wood of these species are economically strategic for the sustainable development and well-being of local communities. At least 674 tree species (23% of the total) contribute to people with at least one material use (e.g., energy, food, materials, medicinal, genetic resources, and others) or non-material use (e.g., learning and inspiration, physical and psychological experiences, and supporting identities) [3,5]. In addition, in many rural communities of Mexico with high margination rates, these tree species are also considered key in poverty attention and improving food security [4,6]. For all these characteristics, the germplasm of these valuable and strategic species should be conserved.

One of the best methods for ex situ conservation of germplasm is to have seeds in long-term storage in a seed bank (SB) [7], which can provide a means of preserving genetic

diversity in a small space and at a minimal cost. SBs provide an efficient component of integrated plant conservation strategies [8–10].

The seed conservation approach in SB applies to both wild and economically important species. While in the first case it is a short-term storage for production purposes such as forestry, in the second, it is for long-term storage to preserve biodiversity. In both cases, the collection of seeds requires an economic analysis as part of strategic planning, which allows for recognizing the social and environmental importance and justifying the necessary resources for its management and sustainability. Due to it being very difficult to assign an economic value to wild species, market price information for some commercial species is used as a basis.

The SB protects valuable biological material in both environmental and economic aspects; in addition, they are considered of strategic importance by contributing to the conservation of threatened species under risk from adverse factors such as climate change and biodiversity loss [11,12]. On the other hand, SBs protect key species, which contributes to market diversification (e.g., new markets with forest species with unique characteristics, such as resistance to pests, higher production, adaptability to climate change, and conservation of biodiversity, others) [13,14].

In Mexico, several studies have emphasized the importance of SBs in maintaining genetic diversity and their strategic role in natural resource conservation [15,16]. In this sense, SBs are valuable public goods that do not have a defined market price [16] although they do have value [17,18]. At the same time, their maintenance and operation in preserving seeds need constant financing resources [16,18,19].

A first approximation to recognize the monetary value of an SB of native species is the elaboration of an indicator that integrates environmental and socioeconomic variables, such as those considered in the Sustainable Development Goals (SDG), which are important for the attention of poverty and food safety [20,21]. This indicator has never been applied in Mexico to analyze the economic importance of SBs in the conservation of native species.

The economic valuation of the fixed and variable costs of seed collection and storage in an SB must be weighed against the importance it represents for society. For this purpose, the problem that we faced was to find a substitute or a simulated market for addressing the financial needs of society [22]. Since there is no direct market price that estimates the value of wild plant diversity or its conservation, it was necessary to adapt some approaches to estimate its social value. In this sense, the approach of environmental and natural resource economics was the ideal framework to evaluate SB collections [22]. There are various methods for economically valuing the conservation of natural resources [22,23] including willingness to pay at market prices, contingent valuation, hedonic prices, as well as the reduction or loss of resources due to their use (depreciation), costs of collection and damage function, among others [14,22–24]. There are economic example assessments of germplasm in the development of a Colombian coffee variety [15] and in the assessment of genetic resources in bean and palm varieties in Costa Rica [25]. It was determined in these studies that the annual cost of operating the germplasm bank is considerably lower than the profits that producers obtain by using the higher-yield varieties stored in the bank. Likewise, the analyses of the operating costs of five SBs of the Consultative Group on International Agriculture (CGIAR) [26] and SBs of Crop Research for the Semi-Arid Tropic, India [27] conclude the need for creating a long-term universal trust and the importance of estimating the operating costs of an SB to formulate profitable conservation approaches, respectively.

Other studies consider that the investment made in conserving phylogenetic resources is a relatively modest sum compared to the enormous benefits that could be generated (e.g., seeds of species with high productivity for feeding people or sustaining world agriculture and the food supply). Furthermore, assessing the costs of germplasm conservation is useful for developing countries, facilitating the appropriate allocation of financial resources and the calculation of economic benefits [14,18,27–30]. The lack of economic assessment of the seeds in a bank can cause financial loss for recouping the investment or insurance payments, due to their loss during transportation to other locations or unexpected accidents

during manipulation, storage, and management of accessions outside the SB. It also helps to estimate the opportunity costs due to the undervaluation of projects for the operation of the SB itself and the underestimation of the system when facing seed scarcity due to unfavorable climate conditions. An SB itself becomes an element that is difficult to evaluate in terms of market price [20]. Ignoring these important arguments leads to inappropriate public and private policies for the ex situ conservation of natural resources.

In order to analyze the costs of collection and ex situ conservation of seeds of some important tree species in Mexico, we established two objectives: (1) Construct a tree index to identify and prioritize native species with economic and social importance, with the potential to reduce poverty and improve food security; (2) carry out an economic evaluation of the collection of seeds and their storage in an SB of those species identified in (1).

This is a pioneering study in Mexico that attempts to address an integrated approach that includes the environmental importance of native species; the relevance of the SBs; the economic value of tree seed storage to support the design and development of public policies that can improve the financial operation of seed banks for conservation purposes. Marginally, the tree index can serve as a tool for researchers and decision makers to prioritize species that, due to their importance to poor communities in the country and environmental relevance, require special attention for their protection or promotion.

## 2. Materials and Methods

We obtained information on the quotation and prices of seeds collected from some forest species by forestry technical service providers and public officials of the National Forestry Commission in Mexico. Likewise, researchers from the SB of the National Autonomous University in Mexico City (hereafter, FESI-SB) were interviewed about the operation and its seed collection according to [3]. The FESI-SB was formally born in 2003, but since 1992, it has collected various types of seeds for research purposes; it is the heritage of the National Autonomous University of Mexico (UNAM); it has no commercial interest and its only objective is conservation and scientific research. It has a collection of 560 accessions and currently does not have an economic valuation in its operations [3,31]. For the costs of collecting and storing seeds, we consulted the work of Rodríguez-Zúñiga et al. [29] and information from FESI-SB researchers. In order to compare the collection costs and their valuation form versus the FESI-SB collection costs of those seeds of tree species with the highest value according to the constructed index, we consulted the "technical forestry service providers" (Prestadores de Servicios Técnicos Forestales—PSTF, based on Spanish initials—who are dedicated to the sale of seeds) [32]. Finally, we used a 10-year seed storage planning period with a nominal interest rate (NIR) of 7%. Extending the period beyond ten years would be less reasonable because it is difficult to predict variables that influence NIR (e.g., expected inflation). The NIR has been used in the last seven years and has been recommended by the Bank of Mexico [33] for planning periods of one year in productive projects.

The study consisted of two stages: (1) construction of an indicator of economic and social importance through multicriteria analysis.

Table 1 summarizes the environmental and socioeconomic variables used in the construction of the indicator that prioritizes tree species. The value of the independent variables was based on: (a) the expert assessment of the variables (*Va*): environmental and social importance, conservation status, food security, and importance of poverty care; (b) native tree species with categories of the *Va* according to Appendices 3, 4, and 15 indicated by [4]; Appendix 4 indicated by [28].

The multicriteria analysis allowed for the numerical assessment of the tree species that were either present or absent based on their environmental and social importance, conservation status, attention to poverty, and importance for food security. These variables were evaluated based on the experience and opinion of ten specialists in natural resource management.

**Table 1.** Environmental and socioeconomic variables used in the indicator construction.

| Name | Variable (*Va*) | Source |
|---|---|---|
| Environmental and social importance | (a) (independent) | Team of experts, FESI-SB, [4] (Appendix 3) |
| Conservation status | (b) (independent) | Team of experts, FESI-SB, [34] (Appendix 4) |
| Importance of poverty attention | (c) (independent) | Team of experts, FESI-SB; [4] (Appendix 15) |
| Food security | (d) (independent) | Team of experts, FESI-SB [4] (Appendix 15) |
| Value of species (*Vse*) | Dependent | |

The criteria used for evaluating these variables were: (1) importance order (1 to 4, 4 being the most important) in the social-environmental dimension and their importance for the community (1 to 10, 10 being the most important), for the social-economical dimension [35] (Equations (1) and (2)).

$$Yj = \frac{\sum_{i=1}^{n} Xij}{n}, \tag{1}$$

where $Xij$ = score by expert $i$ for indicator $j$ ($1 \leq X \leq 10$); $Yj$ = average score for indicator $j$.

$$Vse = \sum_{j=1}^{m} \alpha j Yj, \tag{2}$$

where $j$ = weight assigned to indicator $j$ ($1 \leq \alpha \leq 4$); $Vse$ = value of species $e$, which has a value ranging from 10 (minimum) to 93 (maximum).

(2) Cost analysis of selected species with major indicator

The biological collection (seeds) cost assessment proposed by Rodríguez-Zúñiga et al. [29] was used in this study. This method consists of evaluating the environmental damage caused by anthropogenic activity, measuring the cost of promoting the increase in pollution levels and the changes due to its increment [23], as has been reported for mangroves [23,24]. For this research, in an SB, the background was the analysis of the cost of collecting and storing seeds of some main species of native trees and their future benefit for their conservation.

In this study, we consider the costs related to the collection and storage of the accessions as a variable. An accession represents a sample of 3000 to 4000 distinct and uniquely identifiable tree seeds, which may belong to a cultivar, an improved line, or a population stored for conservation and use [36]. Currently, 560 accessions of tree seeds are stored in the FESI-SB facilities [3].

Table 2 shows a summary of the costs of collecting tree species seeds in different regions of Mexico and sample treatment and storage according to the manual for seed management in germplasm banks, published by Rodríguez-Zúñiga et al. [29] and Rao et al. [29,36], respectively.

We also consulted the PSTF, in which the prices and pricing methods for seed collection is referred, in particular, for those species with high *Vse*. PSTF offer a small number of seeds of tree species from around the country (about 29 species, they offer at the same prices an accession or a kilogram of seed) at a price that is set based on variable costs (collection costs), fixed costs (infrastructure and organizational costs), and a markup that reflects their profit.

In order to obtain comparable prices (i.e., market prices versus collection costs), and for practical reasons, we carried out economic evaluations for the accessions species that met two criteria: (1) the highest *Vse* indicator and (2) the FESI-SB had previously collected and stored at least one accession. In that sense, only eight accession species had these conditions, and these were evaluated. For these eight accessions, the storage cost for ten years was estimated based on the scenario of a productive project with a nominal interest rate equal to that projected by the Central Bank of Mexico [33]. In addition, for the cost



analysis of the FESI-SB, the same approach was used to estimate the variable cost of the FESI-SB accessions.

**Table 2.** Collection costs per accession in different regions of Mexico.

| Item | Regions | |
|---|---|---|
| | Central | Southern |
| (1) Fieldwork | | |
| Per diem for collecting seeds (one collector) | 13.63 | 16.53 |
| Per diem for collecting seeds (team leader) | 29.76 | 37.20 |
| Meals (three persons) | 26.56 | 33.07 |
| Travel collecting expenses for Mexico region (three persons) | 20.67 | 41.33 |
| Airfare (three persons) | 0.00 | 86.81 |
| Other costs (materials and equipment) | 27.93 | 28.93 |
| Subtotal (1) | 118.55 | 243.88 |
| (2) Seed collection permits and shipments | | |
| Local permits | 10.42 | 12.40 |
| Seed shipment | 0.00 | 10.33 |
| Subtotal (2) | 10.42 | 22.73 |
| TOTAL (1) + (2) | 128.97 | 266.62 |
| (3) Annual cost for the treatment and storage of one accession in an SB | | |
| Subtotal (3) | 28.89 | |

Cost in USD. Source: Rodríguez-Zúñiga et al. [29].

## 3. Results

### 3.1. Construction of Economic and Social Importance Indicator through Multicriteria Analysis

Table 3 shows the results of the multicriteria analysis of the values that the experts assigned to each of the variables. The value obtained for each taxon was called "Value of species e" (*Vse*). The scale used to evaluate the species was normalized to three digits (1 to 100) since the scales for the rest of the variables also ranged from 1 to 100. In that sense, 0 is less important and 100 is more important.

**Table 3.** Multicriteria analysis of binary variables.

| Variable Weight = α(Mean Weight Assigned to Indicator *j*) | $Yj = \frac{\sum_{i=1}^{n} Xij}{n}$ | $\sum_{j=1}^{m} \alpha j Yj$ |
|---|---|---|
| (a) Environmental-social (1.0) | 10.0 | 10.00 |
| (b) Conservation status (2.0) | 8.00 | 16.00 |
| (c) Food security (3.0) | 9.00 | 27.00 |
| (d) Importance for poverty attention (4.0) | 10.00 | 40.00 |
| $Vse = \sum_{j=1}^{m} \alpha j Yj$ | | 93.00 |

Where $Yj$ = mean value for indicator *j*; α = weight assigned to resource *j*; *Vse* = maximum value possible for species e.

The values of the binary variables obtained through the multicriteria analysis by the group of experts (Table 3) show that the highest scores obtained when weighing and evaluating the variables were the importance of attention to poverty and food security, presenting values of 40 and 27, respectively.

The *Vse* is based on the individual values of each species in its variables: environmental and social importance, conservation status, importance of poverty, and food security (Equations (3) and (4)). Only 32 species under the categories of importance on attention

to poverty and food security were in the *Vse* interval (value of the species) from 93 to 27 (Table 4).

$$\begin{aligned} Maximum\ value\ of\ species\ e \\ = f(\text{environmental and social importance, conservation status,} \\ \text{importance of poverty attention and food security}) \end{aligned} \tag{3}$$

$$Vse = X_1 + X_2 + X_3 + X_4 \tag{4}$$

where $X_1$ = environmental and social importance, $X_2$ = conservation status, $X_3$ = food security, and $X_4$ = importance for poverty attention.

**Table 4.** Most important final *Vse* values for multi-use tree species.

| Species | Family | Uses | *Vse* (Variable) |
|---|---|---|---|
| *Cedrela odorata* L. | Meliaceae | 4 | 93 (1,2,3,4) |
| *Rhizophora mangle* L. | Rhizophoraceae | 4 | 93 (1,2,3,4 |
| *Pinus jeffreyi* | Pinaceae | 3 | 93 (1,2,3,4) |
| *Swietenia macrophylla* King | Meliaceae | 1 | 83 (2,3,4) |
| *Sterculia apetala* (Jacq.) H. Karsten. | Malvaceae | 1 | 77 (1,3,4) |
| *Ceiba pentandra* (L.) Gaertn. | Malvaceae | 3 | 77 (1,3,4) |
| *Annona muricata* L. | Annonaceae | 5 | 77 (1,3,4) |
| *Alnus acuminata* L. | Betulaceae | 5 | 77 (1,3,4) |
| *Bixa orellana* L. | Bixaceae | 6 | 77 (1,3,4) |
| *Metopium brownei* (Jacq.) Urb. * | Anacardiaceae | 4 | 77 (1,3,4) |
| *Cordia alliodora* (Ruiz and Pav.) Oken | Boraginaceae | 7 | 77 (1,3,4) |
| *Prosopis juliflora* (Sw.) DC. | Fabaceae | 5 | 77 (1,3,4) |
| *Enterolobium cyclocarpum* (Jacq.) Griseb * | Fabaceae | 3 | 77 (1,3,4) |
| *Juniperus deppeana* Steud. * | Cupressaceae | 4 | 77 (1,3,4) |
| *Tabebuia rosea* (Bertol.) Bertero ex A. DC. * | Bignoniaceae | 5 | 77 (1,3,4) |
| *Gliricidia sepium* (Jacq.) Steud. | Fabaceae | 8 | 77 (1,3,4) |
| *Pinus cembroides* Zucc. * | Pinaceae | 3 | 77 (1,3,4) |
| *Leucaena leucocephala* (Lam.) de Wit * | Fabaceae | 6 | 77 (1,3,4) |
| *Guazuma ulmifolia* Lam. | Malvaceae | 7 | 77 (1,3,4) |
| *Bursera simaruba* (L.) Sarg. | Burseraceae | 5 | 77 (1,3,4) |
| *Eysenhardtia polystachya* (Ortega) Sarg. * | Fabaceae | 4 | 77 (1,3,4) |
| *Cordia dodecandra* A.DC. | Boraginaceae | 5 | 67 (3,4) |
| *Pseudotsuga menziesii* (Mirb.) Franco | Pinaceae | 1 | 56 (2,3) |
| *Prosopis laevigata* (Humb. and Bonpl. ex Willd.) M.C. Johnst. | Fabaceae | 6 | 56 (2,3) |
| *Pinus pseudostrobus* Lindl. * | Pinaceae | 1 | 56 (2,3) |
| *Swietenia humilis* Zucc. | Meliaceae | 1 | 56 (2,3) |
| *Brosimum alicastrum* Sw. | Moraceae | 4 | 56 (2,3) |
| *Pinus montezumae* Lamb. | Pinaceae | 5 | 56 (2,3) |
| *Yucca carnerosana* (Trel.) McKelvey | Asparagaceae | 3 | 27 (3) |
| *Prosopis glandulosa* Torr. | Fabaceae | 2 | 27 (3) |
| *Leucaena esculenta* (Moc. and Sessé ex DC.) Benth. | Fabaceae | 6 | 27 (3) |
| *Yucca schidigera* Roezl ex Ortgies | Asparagaceae | 1 | 27 (3) |

Where 1 = environmental-social, 2 = conservation status, 3 = food security, and 4 = importance for poverty attention. Uses (food, livestock forage, medicine, poison, construction, fuel, environment, and social) are grouped based on the first level of the Economic Botany Data Collection Standard [37]. *: Species selected (eight) for the economic assessment of their seeds (accession).

The family Fabaceae was the most representative group of the *Vse* (28%), and it was the one that mainly contributed to the high diversity of uses with 36.24%.

Figure 1 shows the percent contribution of trees native to Mexico to the diversity of domestic uses.

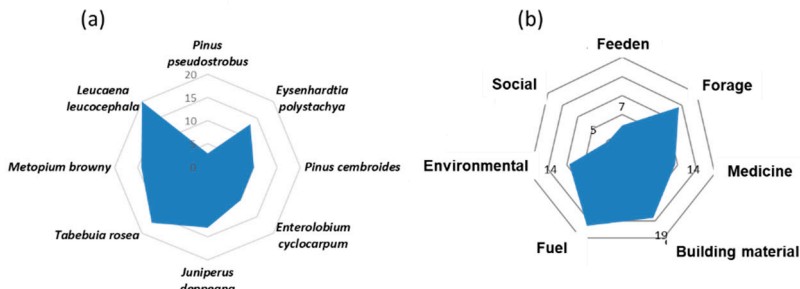

**Figure 1.** Eight focal species with high *Vse* values and which are stored at the FESI Seed Bank. (**a**) Contribution (%) in terms of uses; (**b**) distribution of diversity of uses (%) of the eight focal species selected.

*3.2. Cost Analysis of Selected Species with Major Indicator*

We use the variable costs (salaries, travel expenses, seed storage, etc.) for each of the eight species with high *Vse*, a NIR of 7% recommended for productive projects in Mexico [33], and a depreciation for past inflation. The present (current) and future values of each accession in the FESI-SB can be calculated using Equations (5)–(7) [38], respectively.

$$CCTt = (CCC)/(1 + \pi)^t \tag{5}$$

where *CCTt* = collection costs in the year that the seeds of species "e" were collected; *CCC* = current collection costs of the seed of species "e"; $\pi$ = average inflation in the period: current time and year that the species "e" was collected; *t* = number of years in that period.

$$SCTt = (CSC)/(1 + \pi)^t \tag{6}$$

where *SCTt* = storage costs in the year that the seeds of species "e" were stored; *CSC* = current storage costs of the seed of species "e"; $\pi$ = average inflation in the period: current time and year that the species "e" was collected; *t* = number of years in that period.

$$VAi10 = (CCTt + SCTt) * (1 + i)^{t+10} \tag{7}$$

where *VAi10* = value of accession *i* to 10 years; *CCTt* = collection costs in the year that the seeds of species "e" were collected; *SCTt* = storage costs in the year that the seeds of species "e" were collected; *i* = average interest rate during period *t* + 10 (=0.07); *t* = period between the present time and the year in which the seeds of species "e" were collected and stored.

Table 5 shows a summary of the costs at present (*CCA*) for a planning period of 10 years (*VAe10*) of seed storage of eight tree species with high *Vse*, and Table 6 summarizes the market prices of some seeds compared to their collection and storage costs.

**Table 5.** Accessions of useful forest species with high importance stored at the FESI-SB.

| Species | R | CCC (USD) | ST (years) | CCTt (USD) | SCTt (USD) | VAi10 (USD) |
|---|---|---|---|---|---|---|
| *Pinus pseudostrobus* Lindl. | C | 128.97 | 0.7 | 124.33 | 26.16 | 310.39 |
| *Eysenhardtia polystachya* (Ortega) Sarg. | C | 128.97 | 13.0 | 75.36 | 15.85 | 432.38 |
| *Pinus cembroides* Zucc. | C | 128.97 | 9.2 | 88.33 | 18.58 | 770.95 |
| *Enterolobium cyclocarpum* (Jacq.) Griseb | S | 266.62 | 17.0 | 132.04 | 13.44 | 903.99 |
| *Juniperus deppeana* Steud. | C | 128.97 | 12.1 | 71.46 | 15.04 | 385.83 |
| *Tabebuia rosea* (Bertol.) Bertero ex A.DC. | C | 128.97 | 12.6 | 76.61 | 16.12 | 427.85 |
| *Metopium brownei* (Jacq.) Urb. | C | 128.97 | 17.6 | 62.31 | 13.11 | 488.07 |
| *Leucaena leucocephala* (Lam.) de Wit | C | 128.97 | 17.8 | 61.80 | 13.00 | 490.65 |

Where R = region (C = Central, S = Southern); *CCC* = current collection cost; ST = storage time of the species "e", *CCTt* = collection costs in the year that the seeds of species "e" were collected; *SCTt* = storage costs in the year that the seeds of species "e" were stored; CCA = current cost of the accession stored (present value); *VAi10* = value of accession of the species "e" to 10 years. Source: Author-compiled based on information from the FESI-SB.

**Table 6.** Market price of seeds versus collection and storage costs in the FESI-SB.

| Species | * Market Price in 2018 (USD/Accession) | Current Cost of Seeds Stored at the FESI-SB (USD/Accession) | Current Value for a (100%) Markup of Stored Seeds (USD) | Value of Accession of the Species to 10 Years with Markup (100%)(USD) |
|---|---|---|---|---|
| *Pinus pseudostrobus* Lindl. | 34.72 | 158.33 | 316.00 | 620.78 |
| *Eysenhardtia polystachya* (Ortega) Sarg. | 74.40 | 219.80 | 439.60 | 864.76 |
| *Pinus cembroides* Zucc. | 27.28 | 199.23 | 398.46 | 1541.90 |
| *Enterolobium cyclocarpum* (Jacq.) Griseb | 47.12 | 459.56 | 919.12 | 1807.98 |
| *Juniperus deppeana* Steud. | 29.76 | 196.15 | 392.30 | 771.66 |
| *Tabebuia rosea* (Bertol.) Bertero ex A.DC. | 47.12 | 217.51 | 435.02 | 855.70 |
| *Metopium brownei* (Jacq.) Urb. | 50.13 | 248.10 | 496.2 | 976.14 |
| *Leucaena leucocephala* (Lam.) de Wit | 47.12 | 249.41 | 498.82 | 981.30 |
| Average | 44.71 | 243.51 | 486.94 | 1052.53 |

Source: Author-generated based on information from the FESI-SB, provided by the Forest Restoration Agency (CONAFOR) and technical forestry service providers (PSTF). * PSTF sell the seeds either by a kilogram or an accession.

There are two collection values, according to the spatial distribution of the species: southern regions and the central region. Providers increase the collection costs by 100% to ensure the profitability of their investment and cover the costs of the actions and materials required (fixed capital). Many organizations when facing difficulties calculate fixed capital by using a markup (100%), through which they estimate the fixed costs to produce a good and obtain their profit [38,39]. Taking these facts into consideration, the value per accession of the FESI-SB can be calculated as shown in Table 6, where the value of the seeds stored is nearly 11 times the current market price (accessions stored for 12 years on average), increasing to 24 times over the 10-year storage period.

## 4. Discussion

It is possible to build an index of importance of trees that considers the economic-social and environmental variables, but to improve it in future works, correlate the variables conservation, use, and spatial distribution of the 2885 species of native trees in Mexico. Likewise, prioritize native tree species based on the parameters of taxonomic uniqueness and degree of endemism [40]. This will allow the creation of an index of general importance [3]. On the other hand, consider all the species that do not have the categories of importance on attention to poverty and food security (this is one of the weaknesses of the index built in this work). The definition of the importance indicator and the economic assessment of the accessions at the seed bank determined in this work are essential tools for planning work and actions related to the conservation and sustainable management of tree species biodiversity in Mexico. That is, the indicator will allow prioritizing in situ and ex situ conservation actions of tree species that are important for communities and the preservation of biodiversity in Mexico. Finally, a strength of this index is that, according to the literature review, it considers the main indicators of native trees in Mexico. In this sense, the ranking obtained in this work of forest species is closer to reality in terms of social, environmental, and economic importance.

Regarding the indicator obtained, it is important to mention that it is similar to the one proposed by Koshy et al. [41], who recommended a particular method for making decisions in relation to the priorities for the conservation of tree species. By this means, those species with the higher indicator (the sum of the values of the different variables) are those that should be considered very relevant because of the conservation and importance social points of view.

From all the variables proposed and analyzed in the multicriteria analysis, the highest scores obtained were for importance for poverty attention and food security. These results are similar to those reported by Barrantes et al. [42], who assessed the environmental impact of work in a forest reserve. Of the four variables evaluated, food security and importance

for poverty attention reached the highest scores. These results can also be interpreted similarly to those of Sunderlin et al. [43] who, by using spatial analysis of trees and human communities in seven countries, found that there was a strong positive correlation between forest cover and poor communities. Indeed, studies in Mexico, such as [44], indicated that approximately 370,000 inhabitants of the Tarahumara mountain range depend on forest resources, and that those resources are a part of their subsistence strategy. In addition, the author Bray [45] concluded that when forestry resources do not show significant degrees of perturbation, the local indigenous populations do not need to emigrate because forests are the means of providing food security. For all of these reasons, food security and importance for poverty attention are priority variables in the government's strategic plan for the poorest parts of the country [4,6].

Most of the species with the highest indicator belong to the Fabaceae family, which coincides with the information reported by Rodríguez-Arévalo et al. [29], for the Tehuacan-Cucatlán NPA, and with studies of trees in Mexico by Villaseñor et al. and Beech et al. [1,46] about trees important to society.

Even though the construction of the indicator had the goal of selecting the important tree species and subsequently assessing the economics of storing their seeds, the indicator could also serve as a spatial planning tool for the management of forest species that are important to rural communities. Considering that tree species were evaluated with a focus on social and economic values, and that the financial resources available for specific programs for the conservation of forest genetic resources are limited, it is necessary to consider which of the priority species are critical and more necessary to protect and justify their conservation interventions and actions.

The results of this study show that the value of the seeds is 11 times greater than their current market value, which can be explained by the methods used for their collection and conservation. The detailed work undertaken by the SB is a complex system [27] which requires more infrastructure, facility maintenance, capable personnel, and the development of detailed studies of biodiversity and seed viability. On the other hand, seed sales are designed to maximize profit and decrease costs, prioritizing the volume collected rather than the quality of the biological material. At the same time, if a 7% annual interest rate is considered, as recommended by [33], the value of the seeds over 10 years of storage will be 24 times their market value. The depreciation of the value of seeds stored in the FES-SB due to loss of viability is ruled out because the major costs of an SB are the periodic tests of germination and maintenance of the facilities [47]. These costs guarantee that viable seeds avoid genetic erosion caused by climate change problems, natural causes, and anthropogenic activities. Even more, in 10 years, the seed bank conserves the genetic biodiversity of tree species considered of great importance for the Mexican population: food security and importance for poverty attention. On the other hand, a scarcity rate is not considered since one of the main problems of forest land that Mexico faces is the change in land use and native trees on these lands [48].

The interest rate used in seed storage is a parameter that has been applied in previous reports on the management and conservation of natural resources, such as an interest rate of 5% used by the International Center for the Improvement of Corn and Wheat [49]; an 11% rate for the improvement and storage of commercial forest seeds by the United States Government's National System for the Conservation of Phytogenetic Resources [28]. Another approach can be used to show the costs of the seeds stored in an SB when it has been charged 5.6% of the profits from storing the germplasm of fruit trees [50]. Li et al. and Rodríguez-Zúñiga et al. [18,29] point out that the interest rate for ex situ conservation should be above 8.0%. In this sense, the value of seeds at 10 years could increase even more. This interest rate (8.0%) is reasonable and its value will depend on two main aspects: (a) the expected inflation in the next ten years of 4.7% [51]; (b) scarcity of the resource due to deforestation of 0.3% loss of forest land [52] where the germplasm of the eight native tree species evaluated currently exists. It will also depend on the adverse expected for scenarios climate change.

According to the FAO report [30], the high interest rate of the United States of America is due to the fact that this country is highly competitive, being one of the top three countries dominating forest production worldwide. Mexico, in contrast, has high tree diversity, but the large majority of species has no market value [45]. This also leads to a lack of economic incentives for the owners of forest land, who exchange it for low-yield agricultural crops. In the long term, they impoverish their lands and, in them, native trees of great importance. The lack of a market for genetic resources not only limits the economic value [4], but also leads to underinvestment in SB operations. This work contributes significantly to a better understanding, from an economic point of view, the importance of an SB as an investment space for the conservation of the germplasm of native trees of Mexico.

In addition, several authors [13,14,27,28,53] consider that any investment made in research and conservation of phytogenetic resources is small and that the economic benefits in terms of profitability (marginal benefits) will always be large compared to the cost of operation of an SB (marginal cost). The results of this study add to the previously described assessments, since species with higher indicators also have economic, social, and environmental values.

Ex situ conservation brings various benefits for the government: preserve the natural heritage in the facing adverse scenarios (forest fires, land use change, and climate change) in relatively small spaces. For society in general and the environment, this is one of the few works that establish a methodology to evaluate forests' ex situ conservation, for which there is an ethical responsibility to ensure natural heritage for new generations (non-use value). For the forestry industry and owners of forest land (both with native trees considered important for food security and importance for poverty attention), this means having germplasm with its own characteristics (adaptability and performance) that maximizes profits (use value). The constructed index can also be used for in situ use and conservation, especially for those poor forest communities in the country. The sale of seeds from those trees with the highest index of importance can have an added value in the market for ecological restoration or agroforestry. Experiences in Brazil indicate that sociocultural groups less integrated with the market (e.g., indigenous people) achieve better livelihood results through participation in the seed market [54]. While in the Sahel (Africa), with high levels of community participation, the species best adapted to the environment and of greater economic importance were prioritized; subsequently, quality seeds were collected and seedlings were produced to restore degraded soils [55].

In situ and ex situ conservation represent an option for society, government, and forest owners to manage or conserve genetic resources, while also allowing entrepreneurs to make a profit. It is important to consider that this study does not include the monetary measurement of welfare gains or losses generated by ex situ conservation in the FESI-SB. However, it represents an important component of the system in the analysis of seed collection and storage costs. The elaboration of an index of importance of trees and the analysis of costs of collection and storage of seeds facilitate the estimation of the economic benefits of externalities. This is a background for future research and an essential approach to assess the benefit-cost of this public good.

This study shares important information for future research investment assessments in seed conservation and its potential applications. For example, by using the contingent valuation method, which includes the willingness to pay for a common good, it is possible to estimate scarcity levels of species with a high indicator value, or the profitability of investment in seed storage, through the net value of stored accessions. Further studies should investigate the effects of alternative future scenarios on the costs of seed conservation through changes in the interest rates or based on scarcity of seed sources, opportunity costs in the current consumption of some species versus the preservation of genetic resources for their use, and increases in the cost of supplies for collection and storage, including fixed costs and the possible effects of climate change, among others.

## 5. Conclusions

By analyzing binary variables, a social and economic importance indicator was constructed for native tree species. Based on this indicator, a total of 32 tree species, mainly from the Fabaceae family, are considered very important for improving livelihoods and supporting food security and nutrition in Mexico.

Considering the costs of collection and management recommended for germplasm conservation, the value per accession at FESI-SB (approximately 12 years of storage and an annual nominal interest rate of 7.0%) is 11 times higher than its current market price. In addition, considering 7.0% annual interest rate, the value increases to 24 times the market price over a 10-year storage period. Finally, applying the criteria used for the economic assessment undertaken in this work, we can conclude that the value of 560 lots of tree seed storage at the FESI-SB is approximately USD 124,950. Despite the costs associated with maintaining and managing the SBs, we highlight their importance and contribution to society.

We estimate and analyze the costs of collecting and storing seeds from native species by using a normal storage yield rate (7%). Future research should focus on obtaining the feasibility economic in situ and ex situ conservation of these species in a scarcity scenario (NIR > 7%). Likewise, it should carry out studies of the profitability of the forest lands that the owners could obtain for the conservation and commercialization of germplasm in seed orchards or forest germplasm production units of native trees while considering environmental and social importance, conservation status, importance of poverty attention, and food security.

Finally, this work may be the background to extend the storage period in the year 2050 with an interest rate greater than or equal to 7.0%. In this sense, future research should study the cost benefit of storing the seeds proposed in this work in the FESI-SB or other SBs in the face of adverse scenarios such as climate change, loss of biodiversity, and others.

**Author Contributions:** Conceptualization, J.R.-Z.; methodology, J.R.-Z. and M.d.J.G.-G.; software, J.R.-Z.; validation, J.R.-Z., M.d.J.G.-G., H.B.-S. and F.S.-G.; formal analysis, J.R.-Z.; investigation, J.R.-Z.; resources, H.B.-S.; data curation, J.R.-Z.; writing—original draft preparation, J.R.-Z.; writing—review and editing, J.R.-Z., M.d.J.G.-G., H.B.-S. and F.S.-G.; visualization, F.S.-G.; supervision, J.R.-Z. and M.d.J.G.-G.; project administration, H.B.-S.; funding acquisition, H.B.-S. All authors have read and agreed to the published version of the manuscript.

**Funding:** This research received no external funding.

**Institutional Review Board Statement:** Not applicable for studies not involving humans or animals.

**Data Availability Statement:** Not applicable.

**Acknowledgments:** This paper was published thanks to the support of the COVEICYDET (Consejo Veracruzano de Investigación Científica y Desarrollo Tecnológico). We thank FESI-SB researchers for the valuable information provided.

**Conflicts of Interest:** The authors declare no conflict of interest.

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
