# Peer review of "Cost Analysis of Collection and Ex Situ Conservation of Seeds of Important Native Tree Species of Mexico"

_forests, doi:10.3390/f13111958_

Round 1

Reviewer 1 Report

The manuscript is interesting and develops or adapts an econometric tool to assess seed collections of forest species. It provides quantitative information to justify the conservation of seed banks of tree species that are important for Mexico and its rural communities. 

Comment:  

Line 30.- “The country bears 22 314 plant species and occupies fifth place in the world”. Some text is missing.

Line 37.- What does “material or non-material use” mean. According who..

Line 44.- It is nor clear what “genetic damage” mean. Do you mean inbreeding ?? genetic erosion?

Line 46-68- This phrase is no correct. You can store seeds in the long-terms for both wild and crop genetic resources. The only condition is seed must tolerate dry and later store at very cold condition. Its mean shows orthodox storage behaviour.

Line 85 and 120-121.- It is not clear the what parameters the author used to estimate “annual cost of operating the germplasm bank”. A table similar that was built for estimate collection cost (Table 2) is need for estimate storage cost.

Line 110.- I guess the following phrase in not part of the text of the Methodology “The Materials and Methods should be described with sufficient details to allow others…”

Line 175-176.- Table 2. I think the collection cost is underestimated. Need to include the salary cost per person per day.  Where are the accommodations cost?, What does per diem include?. The table include per diem cost for two peoples but airfare, meals and travelling cost are for three, Why?.  Does traveling cost include 4WD depreciation cost and maintenance?

Table 4.- Columns of uses “a” to “h”, some of them are in capital and other in small capital letters. Why is the reason. Why is relevant to give detail on uses. What It is import the number of uses or type of uses?. If the number of uses per species is in the calculation of dependence variable, could be better summary the uses columns in just one that include the number of uses. Other point, I assume that in the first columns is the result of calculation (Vse). In this case the results go after following columns. The table 4 need some changes for improving. The first columns should be species, later Family, one column of number of uses and finally Vse, the result.

Figure 1.- Axis “y” what does “domestic use” mean. Are there some non-domestic uses?. Axis “x” for “Number of families” you mean number of species per family?. It is confuse !. This figure seem not relevant

Figure 2.- Which are the criteria used for selecting the eight focal species?.  

Table 5.- I understand that Region (R column) affects the collections cost, but I am not clear why SCTt and CCA vary between species. What dependence variables affect the cost of SCTt and CCA?.

Line 263. Table 5.- “SCTt= storage costs in the year that the seeds of species "e" were collected”. It is not clear how the authors obtain SCTt values for each species. Why this value vary between species?. May be the author need to include a Table with details on storage costs. How do you calculate the storage cost?. Did you include the processing cost and monitoring cost?

In Table 2 you did give detailed cost for three regions of Mexico, but later this category practically disappeared from the result and discussion. In Table 5 most species are from Central Region and just one species from South but no one from North region.  Region affects the collection cost, but it does affect the conservation cost?. Please see also comment above.

Line 283.-  The the following phrase: “It is possible to build an index of importance of trees..” you mean “Seed” Index. That wright ? or a more general tree index?

Line 338-339: Phrase confuse. “Many of the seeds contained in the accessions will lose their viability in ten years compared to their current state, which would mean that its estimated value would be depreciated”. It is not clear in what is the “current stage”. Why seed of the species include in this manuscript will lose their viability?. For “current state” you mean at FESI-SB. Need clarification.

Line 377-378.- Following paragraph: “In situ conservation brings various benefits: for the government: preserve the natural heritage in the face of adverse scenarios (forest fires, land use change and climate change) in relatively small spaces.” I think you mean EX SITU !!

Read for introducion/discussion:

https://link.springer.com/article/10.1007/s40003-012-0029-3

https://www.sciencedirect.com/science/article/abs/pii/S0169515003000562

Author Response

The specific response to reviewer 1 comments are found in the attached document

Reviewer 2 Report

The economic evaluation of ex situ conservation activities, in this case tree seed collections, is certainly of great import and use. In the manuscript, the authors create an indicator for the environmental and societal importance of seed collection in a variety of tree species in Mexico. The authors also calculate collection costs for certain collections in Mexico.

While further refining both cost analysis of collections and prioritization tools is of great value, the manuscript as written is confusing and difficult to follow. It appears to need a major rewrite in which the main goals of the work are made more clear, which would probably benefit from consideration of the primary audience for the work. In general, cost analysis and prioritization tools studies are either for the main purpose of 1) presenting a new analysis tool for use by conservation practitioners to inform their collections strategies; or 2) presenting the costs and benefits of germplasm collection to justify a policy response (ie, more or less investment for continuing collections activities). It is not clear from the manuscript who the authors are trying to talk to and what end they hope to achieve.

Also, the authors state in the abstract that “there are few or no studies that estimate the cost of germplasm conservation…” This does not appear accurate. There are many studies, white papers and other documents relating to the costs of germplasm conservation all over the world. The CGIAR is very active in this area (see https://onlinelibrary.wiley.com/doi/abs/10.1111/j.1574-0862.2003.tb00165.x), as are many national government that maintain robust seed-based germplasm collections. A more robust review of germplasm collection cost analyses beyond trees, may help focus the manuscript

There are areas where the paper is hard to follow. For example, on line 158, the authors state that the cost analysis was based on a “…collection (seeds) cost assessment…” in reference 22. However, reference 22 is a paper on contingent valuation of environmental goods, and contains no specific reference to seeds or other biological collections. While the cited paper is an important piece on non-market valuation (which is certainly related to valuing a seed collection that may have market and non-market values), it does not appear clear to me how this information provides a base for the cost analysis in this manuscript. It well may, but it was very difficult to follow.

My suggestion is that the authors reevaluate the presentation of this study with a clear understanding of who their intended audience is. With a clear idea of the intended audience, the manuscript should be rewritten to bring the reader, step by step, through the creation of the authors’ analytical tools and their intended value to the conservation community. Such a rewrite would help focus the writing and allow for a more constructive evaluation of the manuscripts suitability for publication. 

Author Response

The specific response to reviewer 2 comments are found in the attached document

Round 2

Reviewer 1 Report

Row 305-306. References Needed https://www.gob.mx/tramites/ficha/inscripcion-en-el-registro-forestal-nacional-como-prestador-de-servicios-tecnicos-forestales-o-auditor-tecnico-forestal/SEMARNAT1185

Row 549. Need Improve discussion on the economic value conserving the Biodiversity of Mexican Tree (Genetic Resources), in terms of direct, indirect value and Option  Value.

a) For conservation Index: http://www.biouls.cl/lrojo/lrojo03/public_html/libro/19.pdf

b) For socio-economic assesment and role of local communities: https://www.jstor.org/stable/26309563

https://onlinelibrary.wiley.com/doi/full/10.1111/rec.12337

https://www.pnas.org/doi/10.1073/pnas.0508036102

https://link.springer.com/article/10.1007/s10640-022-00674-1

https://www.sciencedirect.com/science/article/pii/S0006320721003268
